# Translation of culturally and contextually informed diabetes training for Aboriginal primary health care providers on Aboriginal client outcomes: Protocol of a cluster randomized crossover trial of effectiveness

**Odette Pearson**[1,2]*, **Sana Ishaque**[1,2], **Saravana Kumar**[3], **David Jesudason**[2,4], **Paul Zimmet**[5], **Gloria C. Mejia**[6], **Gary Wittert**[2,7], **Sara Jones**[8], **Jane Giles**[9], **Natalie Wischer**[10], **Sophia Zoungas**[11], **Sarah Crossing**[1,2], **Sarah Davey**[12], **Tinarra Toohey**[1,2], **Satinder Kaur**[13], **Alex Brown**[1,2,14,15], **Tina Brodie**[1,2], **Shwikar Othman**[1,2], **Kim Morey**[1,2]

1 Wardliparingga Aboriginal Health Equity, South Australian Health and Medical Research Institute, Adelaide, South Australia, Australia, 2 Adelaide Medical School, Faculty of Health and Medical Sciences, University of Adelaide, Adelaide, South Australia, Australia, 3 Allied Health and Health Services Research, School of Allied Health and Human Performance, University of South Australia, Adelaide, South Australia, Australia, 4 Endocrinology & Diabetes Department, Queen Elizabeth Hospital, Central Adelaide Local Health Network, SA Health, Adelaide, South Australia, Australia, 5 Department of Diabetes, Central Clinical School, Monash University, Melbourne, Victoria, Australia, 6 Australian Research Centre for Population Oral Health (ARCPOH), University of Adelaide, Adelaide, South Australia, Australia, 7 Freemasons Centre for Male Health and Well Being, South Australian Health and Medical Research Institute, Adelaide, South Australia, Australia, 8 Rural Health Education and Training, Allied Health and Human Performance, University of South Australia, Adelaide, South Australia, Australia, 9 Diabetes Service, Rural Support Service, Regional LHNs | SA Health, Adelaide, South Australia, Australia, 10 National Association of Diabetes Centres, Sydney, New South Wales, Australia, 11 Epidemiology and Preventive Medicine, School of Public Health and Preventive Medicine, Monash University, Melbourne, Victoria, Australia, 12 Aboriginal Health Council of South Australia Ltd, Adelaide, SA, Australia, 13 Diabetes SA, The Diabetic Association of South Australia Incorporated, SA Health, Adelaide, South Australia, Australia, 14 Indigenous Genomics, Telethon Kids Institute, Adelaide, SA, Australia, 15 Indigenous Genomics, Australian National University, Canberra, ACT, Australia

* Odette.Pearson@sahmri.com

## Abstract

### Background

Indigenous populations globally have significantly high rates of type 2 diabetes compared to their non-Indigenous counterparts. This study aims to implement and evaluate the effectiveness of a culturally and contextually informed Aboriginal Diabetes Workforce Training Program on Aboriginal primary health care workforce knowledge, attitude, confidence, skill and practice relating to diabetes care.

### Methods

A Cluster Randomised Crossover Control Trial with two arms (Group A and Group B) will be conducted with Aboriginal primary health care services in South Australia. These services primarily provide primary health care to Aboriginal and Torres Strait Islander people. All

**Data Availability Statement:** No datasets were generated or analyzed during the current study. In accordance with ethical restrictions, details for requesting access to the relevant data will be provided upon study completion.

**Funding:** This research is Funded by National Health and Medical Research Council - Medical Research Future Fund (MRFF) Primary Health Care Research Initiative (PHCRI) [APP1200314]. The sponsor does not have any role in the study design, data collection and analysis, decision to publish, or preparation of the manuscript.

**Competing interests:** Odette Pearson, Alex Brown, David Jesudason, Paul Zimmet, Saravana Kumar, Gloria Mejia, Gary Wittert, Sara Jones, Jane Giles, Natalie Wischer, Sophia Zoungas, and Kim Morey received fund from National Health and Medical Research Council - Medical Research Future Fund (MRFF) Primary Health Care Research Initiative (PHCRI) [APP1200314]. The sponsor does not have any role in the study design, data collection and analysis, decision to publish, or preparation of the manuscript.

healthcare service sites will be randomised into groups A and B to receive the training program. The training program consists of three components: 1) Peer support network, 2) E-Learning modules and 3) onsite support. Aboriginal Health Workers of participating sites will be invited to participate in the monthly online peer support network and all chronic disease staff are eligible to participate in the E-Learning modules and onsite support. The Peer Support Network runs for the entirety of the study, 17 months. Training components 2 and 3 occur simultaneously and are 2.5 months in length, with a six-month washout period between the two randomised groups undertaking the training. All primary outcomes of the study relate to diabetes management in a primary health care settings and measure participants' knowledge, attitude, confidence, practice and skills. These will be collected at seven time points across the entire study. Secondary outcomes measure satisfaction of the peer support network using a survey, interviews to understand enablers and barriers to participation, health service systems characteristics through focus groups, and medical record review to ascertain diabetes patients' care received and their clinical outcomes up to 12 months post training intervention.

## Discussion

The findings will explore the effectiveness of the training program on Aboriginal primary health care provider knowledge, attitude, confidence, skill and practice relating to diabetes care. The final findings will be published in 2027.

### Trial registration

The study was prospectively registered in The Australian New Zealand Clinical Trials Registry (ANZCTR), with registration number ACTRN12623000749606 at ANZCTR - Registration. Universal Trial Number (UTN) U1111-1283-5257.

## Introduction

Noncommunicable diseases account for 74% of deaths globally, with 4.8% of those deaths attributable to diabetes [1]. Well controlled diabetes is associated with increased longevity and prevention or slowing the progression of diabetes-related complications. A prospective observational study reported that each 1% reduction in mean HbA1c was associated with reductions in risk of 21% for any end point related to diabetes (95% confidence interval 17% to 24%, $P < 0.0001$), 21% for deaths related to diabetes (15% to 27%, $P < 0.0001$), 14% for myocardial infarction (8% to 21%, $P < 0.0001$) [2], and up to a 50% reduction in the risk of microvascular complications and cardiovascular events [3]. However, the translation of evidence-based guidelines in real-world settings do not achieve the outcomes seen in highly regulated clinical trial settings.

Indigenous populations globally have significantly high rates of type 2 diabetes compared to their non-Indigenous counterparts. In Australia, the Aboriginal and Torres Strait Islander population is no exception. For many, chronic disease risk factors present early in life and lead to type 2 diabetes in early adulthood. Type 2 diabetes is seven times more likely in Aboriginal and Torres Strait Islander adults compared to non-Indigenous adults [4] and responsible for a large proportion of premature morbidity and mortality. Of those who have diabetes, many live

with a high burden of diabetes complications of the heart and blood vessels, kidneys, eyes and nervous system [5]. Diabetes and its associated complications are directly attributable to high mortality rates, with Aboriginal and Torres Strait Islander adults being six times more likely to die from diabetes than non-Indigenous Australians [4].

With diabetes primarily managed within the community setting by multiple disciplines, it is essential that local primary health care services have a workforce that can deliver a minimum standard of diabetes care. Public health evidence across low, middle and high-income countries demonstrates that community health workers are effective in improving population health, particularly where health workforce resources are limited and where health disparities persistent despite well-developed health systems [6, 7]. In Australia, examples include, Aboriginal health workers managing diabetes through case management [8] and continuous quality improvement [9] in primary care settings achieving a 1% and 0.4% reduction in HbA1c, respectively. Investing in the remote area Aboriginal Health Worker and Practitioner (AHW/Ps) workforce can reduce the risk of diabetes-related hospitalisations and can result in hospital cost savings [10].

Diabetes is a specialty field of health care practice. Currently, there is no nationally recognised diabetes course that advances AHW/P knowledge and skills in diabetes. AHW/P in primary care services have completed either a Certificate 3 or 4 in Aboriginal Primary Health Care which is delivered by a Registered Training Organization. The diabetes component of this training is based on raising awareness of diabetes, screening for diabetes and the provision of healthy lifestyle information, all of which is covered over the course of half day. The next level of diabetes training for AHW/P is a Graduate Certificate of Diabetes Education and Management that qualifies as initial credentialing by the Australian Diabetes Educators Association. This is followed by 1,000 hours of training required for full credentialing and on-going training to remain credentialed. While this is a specialised diabetes pathway that some AHW/P would be interested in, there is a gap in the career pathway for those who deliver diabetes care and are seeking additional knowledge and ongoing best practice support, without becoming credentialed diabetes educators. As AHW/Ps are integral to the multidisciplinary diabetes team in Aboriginal primary care services and permanently available and accessible within the community, a comprehensive understanding and practice of diabetes education and management will serve to strengthen the effectiveness of their role.

There is consistent high-level, high-quality evidence to indicate that multifaceted interventions which focus on local contexts, address barriers and promote engagement with health care providers are more likely to result in adoption of best practice and hence improved outcomes [11]. A multifaceted knowledge transfer strategy can have positive impacts, as it considers a number of factors which may influence the uptake and implementation of best practice, including but not limited to knowledge, skills, attitudes, personal beliefs and social and cultural context. A tailored intervention strategy may utilise a combination of educational meetings, training of health care providers, educational outreach, practice facilitation, local health leaders, peer support and audit and feedback [11]. These interventions, individually and collectively, form the building blocks of knowledge transfer and evidence implementation strategies and have been rigorously evaluated. Given that knowledge, skills, attitudes, personal beliefs and social context of the proposed cohort of health care providers are likely to vary, a multifaceted approach will allow for customisation and development of tailored interventions. Such a nuanced approach is critical as barriers to evidence-based diabetic care is also multi-factorial and there is no "one-size fits all" when developing these interventions.

The Aboriginal Diabetes Workforce Study (the current study) involves co-designing, implementing and evaluating an Aboriginal Diabetes Workforce Training Program (training program) with health professionals employed by Aboriginal Primary Health Care Services in

South Australia (SA). This study is a direct response to several workforce strategies within the SA Aboriginal Diabetes Strategy 2017–2021 and that remain a priority based on recent discussions with Aboriginal primary health care services workforce.

During 2021, seven Aboriginal Primary Health Care Services across SA co-designed the training program with the research team, including the study investigators, a are multi-disciplinary team of experts within their field and industry partners. Industry partners included Diabetes Queensland and Diabetes Australia. The development of the E-Learning modules was funded by the National Diabetes Services Scheme (NDSS). The training program consists of three elements: 1) State-wide Peer Support Network (PSN) for AHW/Ps, 2) self-paced E-Learning modules, 3) Onsite Practice Support (OPS).

A cluster randomised control design at the service level will be used to evaluate the effectiveness of the training program on health care provider knowledge, confidence, attitude, practice and skill related to diabetes management. Health care providers include Aboriginal Health Workers, Aboriginal Health Practitioners and multidisciplinary health care providers who provide care to patients with diabetes and work in Aboriginal primary health care services across South Australia. Implementing the training program has the potential to embed a minimum standard of diabetes training across the state and professionally develop and support the Aboriginal Health Worker and Practitioner workforce who are vital to managing diabetes in the Aboriginal community.

Thus, the primary aim of this clinical trial is to evaluate the effectiveness of a culturally and contextually informed Aboriginal Diabetes Workforce Training Program (hereafter training program) on Aboriginal primary health care workforce knowledge, attitude, confidence, skill and practice relating to diabetes care.

## Objectives

To evaluate the effectiveness of a training program on Aboriginal Health Workers, Practitioners and multidisciplinary team knowledge, confidence, skill, practice and attitude related to diabetes care, and secondary outcomes relating to quality of diabetes care and patient outcomes.

## Materials and methods

### Study design and setting

The study uses a Cluster Randomised Crossover Control Trial with two arms (Group A and Group B). The study will be implemented in Aboriginal primary health care services in South Australia. These services primarily provide primary health care to Aboriginal and Torres Strait Islander people and are located across metropolitan, regional and remote areas of South Australia. This study will be conducted in South Australia, between 15th February 2024 to 30th September 2026.

This protocol is reported according to the Standard Protocol Items: Recommendations for Interventional Trials (SPIRIT) guidelines (S1 File. SPIRIT Checklist) [12, 13].

### Eligibility criteria

**1. The training program (the intervention).**   Aboriginal Health Workers/Practitioners who are:

∘ qualified or currently enrolled in Certificate III/IV in Indigenous Primary Health Care,

∘ employed by a participating South Australian Aboriginal primary health care service, and

○ provide diabetes care to Aboriginal clients.

Multidisciplinary health care providers who are:

○ from a broad range of service providers, including enrolled and registered nurses, allied health professionals and medical doctors,

○ employed by or provides services in a participating South Australian Aboriginal primary health care service,

○ provide diabetes care to Aboriginal clients.

**2. Interviews for enablers and barriers.** All training program participants are eligible for an interview on the enablers and barriers to participation, effectiveness and sustainability of the training program once they have completed the training program.

**3. Focus group for service systems assessment.** Chronic disease team members in each participating service will be invited to attend a focus group to assess the system characteristics for each service.

**4. Medical record review.** People with diabetes who have attended the participating health service and who fulfil the following criteria at the time of medical record review:

- Both males and females who aged 18 years and over;

- identify as Aboriginal or Torres and Strait Islander people;

- have a confirmed diagnosis of T2DM and/or a HbA1c is greater than or equal to 6.5 mmol/L;

- have attended the health service at least twice in the preceding 12-month period.

## Exclusion criteria

1. Training program: Limited or no access to a stable Internet connection or computer hardware (laptop, desktop, tablet) can be provided by the health service.

2. Training program enablers and barriers interviews: No exclusion criteria.

3. Service systems assessment focus groups: No exclusion criteria.

4. Medical record review: No exclusion criteria.

## Recruitment strategies and sample size

Participants will be recruited according to each area:

**1. Training program.** All participants will be recruited prior to the commencement of the training program. Only staff employed in sites with a collaborative agreement signed by the service CEO (or delegate) will be approached. Recruitment of training program participants will be by indirect approach. A manager of the primary health care service will identify eligible staff based on the eligibility criteria. Primary Health Care service manager or delegated clinic manager will send an email, drafted by the research team to potential participants who meet the inclusion criteria. The email will include an introduction to the training program, the information sheet and the consent form, (S2 and S3 Files. Participants information sheet and Consent form). Participants will be required to contact the research team to flag their interest in participating in the program. Contact details of the Study Manager are included on the information sheet and in the email.

*Sample size calculation and drop-out rate.* The study population will be drawn from primary health settings and will include Aboriginal Health Workers and Practitioners, and multidisciplinary health care providers across South Australia. Based on our scoping review "Supporting best practice in the management of chronic diseases in primary health care settings: a scoping review of training programs for Indigenous Health Workers and Practitioners" currently under review, the review reported improvement in participants' knowledge and confidence of a similar population group between 12.7%-20.97% increase pre-post training, and an increase in confidence in both clinical and non-clinical skills. The sample size for the studies included in this review ranged from 4 to 250 with a total of 1120 participants. The drop-out rate was reported to be between 40 to 70% during the follow up period. Furthermore, based on the publicly available information from the Australian Health Practitioner Regulation Agency (AHPRA), a total of (n = 100) Aboriginal health practitioners are currently registered and licenced to work in health settings across South Australia (this does not include Aboriginal Health Workers). There are 11 Aboriginal community-controlled health services and 10 South Australian Local Health Networks, some of which have expressed interest in participating in the study. Overall, we have estimated an uptake of (n = 40) and accounted for drop-out rate of 50% of the AHW/P, therefore, (n = 20) participants would need to complete the training. This number aligns with previous studies [14–16].

As participation in this project is voluntary, potential participating sites may decline to participate and participants may withdraw from the study. In the event that that a participant wishes to withdraw from the study, they will be asked to complete a participant withdrawal form, and this will be reported in the final findings. A minimum of two participants will be recruited from each study site, with no restriction on the maximum number of participants from each service. Recruitment will include participants from primary health service settings, across urban, regional, and remote locations. We aim to include a minimum of two sites in each setting. Three months has been allocated to recruit participants.

**2. Training program enablers and barriers interviews.** All participants will be advised in the training program information sheet and participant consent form that they will be invited to participate in an interview about the barriers and enablers of the training program after they have completed the training. On completing the training program participants will be directly approached by email by a research team member up to three times over 3 weeks to be invited to participate in an interview. A subsample of training program participants will be recruited through a purposeful sampling approach to achieve diversity in disciplines and healthcare service location - urban, rural and remote. A maximum of 15 participants will be recruited given time and feasibility.

**3. Service system assessment focus groups.** Focus groups with the chronic disease workforce in each participating health services will be conducted to assess the system characteristics for each service. The systems assessment is negotiated in the Collaborative Agreement developed prior to study commencement with each service. Although, services can decide at any point prior to the systems assessment that they do not want to be involved in the system assessment. If a service agrees to participate in the systems assessment, the CEO (or their nominee) will be provided by the Study Manager with a draft email inviting diabetes/chronic disease team members to participate. The participant information sheet will be attached in the email. The CEO will arrange a time for the focus group inviting all of those they identify as eligible. Only those interested in participating will attend and initially hear about the study and the purpose of the systems assessment.

There is no upper or lower limit to the number of staff who participate in the focus group; it depends on the staff willing to participate and who are available on the day.

**4. Medical record review.** Participants will not be recruited; existing data will be audited retrospectively. The number of people with diabetes who receive care at each service enrolled in the study and meet the study inclusion criteria will vary. Prevalence of self-reported diabetes varies between 8 and 23% according to the 2018–19 Australian Bureau of Statistics National Health Survey [17]. It is therefore expected that between 8 and 23% of clients at each enrolled service will meet the eligibility criteria. The number of clients registered with a service varies from 100 through to 3,000.

Medical records of people with diabetes in each participating service who meet the inclusion criteria will be audited. Services will confirm their participation in the medical record review during the development of the Collaborative Agreement. This will be revisited during the first 12 months of the project at the time we are arranging to collect data. Services can decide then or at any point that they do not want to be involved in the medical record review. There is human and financial support to assist with the extraction of data.

## Randomization and assignment of intervention

Randomization and allocation will occur prior to the commencement of the intervention and at the service level. Randomization will be undertaken after six months of providing PSN for all enrolled health service sites. All enrolled health service sites will be listed, and a statistician will be responsible for randomizing the health service sites into cluster one (Group A participants) or cluster two (Group B participants) using a block size through a computer program for randomisation.

For the *intervention assignment*; a cluster crossover randomised trial will be employed in this study, as all participants receive the same intervention based on random service site allocation. Participants in the study sites will know which group they have been allocated after completing six months of the PSN, and before commencing the E-Learning module for group A. The project research team will not be blinded to group A or B allocation, only people analyzing the results will be masked/blinded.

## Implementation

The chief investigator and study manager will recruit participants to the training program, interview, and focus group. The study manager will allocate participants with a unique ID through RedCap once they have consented to participate. After randomization, consenting AHW/Ps from both groups (A & B) will participate in the PSN for six months duration. At the start of the month seven, study sites randomised to Group A, will commence the E-Learning modules and OPS, for a duration of 10 weeks. During this time, AHW/Ps from Group A & B will continue to participate in the PSN. This will be followed by a wash out period of six months for both group A and B. Then Group B will commence the E-Learning modules and OPS, for a duration of 10 weeks.

## Intervention

The Diabetes Training Program consists of three components:

**1. Peer Support Network (PSN).** A state-wide virtual PSN for Aboriginal Health Workers and Practitioners. The PSN will convene for 6 months prior to (Group A) doing the E-Learning modules and the Onsite Practice Support (OPS). The PSN will run online via Teams once a month, for 1.5 to 2 hours duration, and will be facilitated by a credentialled diabetes educator (CDE) employed by the study.

The PSN will include topics aligned with the professional scope of practice that cover diabetes-related content. In addition, activities will be included such as mentorship support,

reflections and readings, stories and lived experiences (S4 File. Peer Support Network Facilitator guide).

**2. E-Learning modules.** There are nine self-paced E-Learning modules. Each module takes approximately an hour to complete. Ten weeks have been allocated to complete the E-Learning modules. All participants are eligible to do the E-Learning modules. The nine self-paced E-Learning modules [18] will cover the following topics related to diabetes care:

- Introduction

- About Diabetes

- Healthy living

- Glucose monitoring

- Low and high blood glucose levels

- Medicines and insulin

- Diabetes-related complications

- Support for self-management

- Other priority groups.

Modes of content delivery within the E-Learning modules include interactive activities and stories, short videos, and links to culturally appropriate resources to support people living with diabetes and their families, with pre- and post-knowledge evaluation.

During the training intervention, a weekly email will be sent out to participants advising of which modules should have been completed to meet the 10-week completion timeframe. Person support will be made available with access to a study team member with clinical experience should participants require. Participants will receive ten professional development points and a certificate on completion of the modules.

**3. Onsite Practice Support (OPS).** Onsite practice support is practical support for staff to assist with implementing knowledge into practice within their primary care service. The OPS sessions will occur during the same 10-week period as the E-Learning modules. The OPS will be delivered in-person at the primary healthcare service.

The number of OPS sessions and what is covered will be negotiated between the participants and the OPS Facilitator. It is suggested that each service hold a minimum of 2 OPS sessions. The OPS sessions will be inclusive of all AHW/P and multidisciplinary healthcare providers enrolled in the study in each health service. The OPS sessions will be held with AHW/Ps for the first half of the session and then include multidisciplinary healthcare providers for the remainder of the session. This format will be assessed and adapted as needed onsite and is in response to direct feedback during the co-design of the training program. The OPS will be facilitated by CDE who is employed by Diabetes SA (S5 File. Onsite Practice Support Facilitator guide).

For the research documentation, the PSN and OS Facilitators will keep a reflective journal that will describe the session from their perspective, barriers and enablers to successful uptake and impact and what could be improved. As the Facilitators are research team members, on advice from the HREC sub-committee, they are not considered research participants and their documentation can be used to inform improvements of the study and be used to describe and evaluate the study. They will record the date of the session and document:

- Detailed description of what was done during each session.

- From the facilitators' perspective, what went well in the session and what could have been done better and potential mitigation or strategies for improvement.

- They will be asked not to record individual or identifying information of individual participants.

These will be collected within 5 days of each session by the Study Manager.

## Washout period

All study sites will have a period of 6 months for washout where PSN is only running (for both group A and B). All participants will be advised in the training program information sheet and participant consent form that they will be invited to participate in an interview about the barriers and enablers of the training program after they have completed the project. On completing the training program participants will be directly approached by email by one of the research team members up to three times of follow up reminder over 3 weeks to be invited to participate in an interview.

## Data collection

**1. Peer support network satisfaction survey.** A peer support network satisfaction evaluation survey will be collected monthly after each peer support network meeting. Using a developmental evaluation approach, data collected from this survey will be used to improve the PSN by responding to positive suggestions and addressing barriers, continuously over the life of the PSN.

**2. Service systems assessment focus groups- Health service characteristics.** Primary health care services vary in the systems and infrastructure available to enable and support the management of type 2 diabetes. The importance of assessing these service characteristics and taking them into consideration in interpreting study findings was emphasized in discussions with the Aboriginal Health Council of South Australia in the development of the grant proposal.

Focus groups with the chronic disease team or staff will be conducted to assess the system characteristics of each service and to understand the service infrastructure, resourcing, and support for best practice chronic disease care. These focus groups will be conducted face-to-face or via Zoom. Focus groups will occur at a time suitable to the service, within 12 months of trial commencement, and will be conducted by a minimum of two research team members. The Assessment of Chronic Care Scale (ACIC) techniques developed by Bonomi and Colleagues [19] will be used to inform the focus groups discussion and system assessment, and will inform data collection and analysis. Participants will be provided with the assessment of Chronic Care Scale prior to the focus group. The focus groups will not be recorded. Staff who participated will be given the opportunity to review the assessment and provide feedback, which will be added.

Only one focus group is proposed for each service (study site), taking approximately 1 hour. In addition, there is no certain number for the size of the focus group; it depends on the staff who are available that day. Participation will be during work hours as agreed with the service manager and participants will not be reimbursed for their participation.

## Following the training program

**1. Interviews for enablers and barriers.** Participants will be interviewed about their participation, sustainability and effectiveness of the training program. Semi-structured interviews will be conducted by an experienced qualitative researcher who is a team member with a

medical background with support from a practicing Aboriginal Health Practitioner. The participants will be provided with the interview questions prior to the interview. Interviews will be conducted face-to-face or via Zoom and will last up to 60 minutes. If face to face, participants will be asked if they could arrange a space within their health service or if they would like the study team to arrange a space in public area such as a room in a local library or in another organization. While all participants will be invited to participate in an interview, a maximum of 15 participants will be recruited given time and feasibility. If there is a choice (based on recruitment), we aim to have representation across geographic locations and service sizes and systems infrastructure, with a mix of AHW/Ps, Aboriginal nurses, and multidisciplinary staff. Participants will be asked for permission to record the interview for transcription and will be given an opportunity to review their transcript to add or remove information.

**2. Medical record review (quality of diabetes care and diabetes clinical outcomes).** Negotiating collaborative agreements with sites prior to study implementation includes the sites agreeing or not agreeing to provide de-identified data from their electronic patient management system (PMS) on their diabetes clients. As the sites do not all use the same type of PMS, the study team will work with the clinic/data manager at each site to determine the best data extraction method for their individual electronic PMS. This will likely require a data extraction report to be written (coded) by the site data manager or by the PMS vendor (e.g., Medical Director, Communicare). Data extraction software may also be utilized, such as CAT Plus from PENCS, which compatible with the majority of PMS across Australia. All identifiable data extracted from a PMS using CAT Plus is securely stored within the service and not moved offsite.

The medical record review will be conducted through an electronic patient information system using automated data extraction algorithm for retrospective data collection at three time points during the study. This was informed by best practice clinical care guidelines–assessed against Royal Australian College of General Practitioners evidence-based care guidelines [20].

The data extraction will be run onsite by an appropriate staff member (as identified by the health service) and saved as a CSV file. The extracted data will be de-identified. The CSV file will be password protected and sent by secure email to the Study Manager. The study team will support the services during data extraction, as required, including providing reimbursement for their time. The number of times electronic medical record data is collected will be negotiated with each individual service - ideally, at the beginning, midway and at the end of the project (namely, at T1, T4 and T6, which will be discussed later in details).

Data will be extracted such as: client ID, month and year of birth, sex, ethnicity, GP management plan, diabetic retinal check performed, diabetic foot check performed, cardiovascular risk assessment, body mass index, waist circumference, smoking status, physical activity, vaccination status, and clinical measurements such as cholesterol (mmol/L), LDL (mmol/L), HDL (mmol/L), triglycerides (mmol/L), creatinine ($\mu$mol/L), eGFR (ml/min/1.73m$^2$), uACR (mg/mmol), random BGL (mmol/L) - non-fasting, blood glucose test performed at the health service, HbA1c (mmol/mol) HbA1c (%), systolic blood pressure (mm/Hg), and diastolic blood pressure (mm/Hg).

## Study outcomes

**Primary outcomes.**   The primary outcomes of this study relate to diabetes management in a primary health care setting that measure participant:

- Knowledge measured using the Simplified Diabetes Knowledge Scale [21].

- Attitude is measured using the Diabetes Attitude Scale (DAS-3) [22].

- Confidence measured using the Diabetes Confidence Survey (DCS) [15].

- Practice and skills using case studies scenarios that were developed by the research team and the investigators specifically for the project and will be assessed as a composite outcome.

- Skill assessed using a case study for a diabetes foot check. This measure was adopted from the Foot Forward Project, which was developed by Twigg, Wischer, and Frank 2021 [23].

**Secondary outcomes.** The secondary outcomes for this study relate to participants' satisfaction with the training program, patient care received and patients' clinical outcomes:

- Peer Support Network satisfaction survey, this survey was developed and informed by a scoping review of the literature on communities of practice.

- Enablers and barriers to participation, effectiveness and sustainability of the training program, this will be collected through semi-structured interviews. The interview questions were informed by scoping review of the literature on chronic disease training programs for Community Health Practitioners.

- Service systems assessment- Health Service Characteristics: System assessment through conducting focus groups with the chronic disease team staff. The Assessment of Chronic Care Scale techniques developed by Bonomi and Colleagues [19] will be used to inform data collection and analysis.

- Quality and Outcomes of diabetes care: This will be conducted through an electronic patient information system, using automated data extraction algorithm.

## Data collection methods and timeline

At each relevant time point of data collection, each participant will receive an email from the Study Manager with a unique URL link to the RedCap Survey. Surveys can be programmed to be sent automatically on a given date which can be changed as well as sent manually if required. For incomplete surveys, up to three email prompts and one phone call will be used to remind participants to complete the survey. Each time a survey is sent, participants will have up to 14 days to complete the survey, then it will be closed.

Participants demographics are collected three times during the study period and will take 5 minutes to complete. The primary outcomes are collected seven times and in total take up to 35 minutes to complete. The peer support network evaluation questions are collected monthly after each PSN session and take approximately 5 minutes to complete (S6 File. Data collection methods).

**Data collection timepoints for primary outcomes.** All primary outcomes of this study relate to diabetes management in a primary health care setting that measure participant knowledge, attitude, confidence, practice and skills will be collected for the entire study for seven points in time. These are labelled T0 (baseline), T1 to T6, and are referred to in the below outline of data collected, (Fig 1: Timeline for recruitment, allocation and assessments).

Timeline for all primary outcomes, with recruitment start date mid-February 2024, as follow:

- T0- Baseline - groups A & B

- T1 - Measure Effect of Peer Support Network - 6 Months Post Baseline Group A & B (Aboriginal Health Workers and Practitioners only).

| | STUDY PERIOD | | | | | | | | | |
|---|---|---|---|---|---|---|---|---|---|---|
| | Enrolment | Allocation | | | | | Post-allocation | | | Close-out |
| TIMEPOINT** | $-t_1$ | 0 | t0 | t1 | t2 | t3 | t4 | t5 | t6 | $t_x$ |
| **ENROLMENT:** | | | | | | | | | | |
| **Eligibility screen** | X | | | | | | | | | |
| **Informed consent** | X | | | | | | | | | |
| **Allocation** | X | X | | | | | | | | |
| **INTERVENTIONS:** | | | | | | | | | | |
| *[Intervention A]* | X | X | ◆—————————————————————————————◆ | | | | | | | |
| *[Washout period]* | | | | | | | | | | |
| *[Intervention B]* | X | X | ◆—————————————————————————————◆ | | | | | | | |
| **ASSESSMENTS:** | | | | | | | | | | |
| *[Diabetes Knowledge]* | | | X | X | X | X | X | X | X | |
| *[Attitude relating to diabetes]* | | | X | X | X | X | X | X | X | |
| *[Confidence in managing diabetes]* | | | X | X | X | X | X | X | X | |
| *[Diabetes Management in Practice and Skills]* | | | X | X | X | X | X | X | X | |
| *[Diabetes Management skills using foot check]* | | | X | X | X | X | X | X | X | |
| *[Enablers and Barriers of the Training]* | | | | | X | | X | | | |
| *[Medical record review]* | | | | X | | | X | | X | |
| *[Peer support network]** | | | | | | | | | | |
| *[Service system assessment]*** | | | | | | | | | | |

* Peer support network will be collected monthly after each PSN

**Service system assessment will be conducted once between month 3 and 12 of the study commencement

**Fig 1. Timeline for recruitment, allocation and assessments.**

- T2 –Measure effect of training program (PSN, Online Modules, Onsite Support) on group A - 8.5 Months post-baseline (data collected on both group A & B).

- T3 –Baseline of group B prior to doing the intervention (data collected on both group A & B), 14.5 months post-baseline.

- T4 –Measure effect of training program (PSN, Online Modules, Onsite Support) on group B post group B undertaking the intervention at 17 months from baseline (data collected on both group A & B).

- T5 –Sustainability of effect of training program on group A 23 months post-baseline, and group B 6 months post commencement of intervention.

- T6 - Sustainability of effect of training program on group A 29 months post-baseline, and group B 12 months post commencement of intervention.

**Data collection timepoints and methods for secondary outcomes.** The secondary outcomes related to the patients care received and patients' clinical outcomes, will be collected through:

1. Peer Support Network satisfaction survey: This will be collected monthly after each Peer Support Network meeting for a total of 29 months post-baseline.

2. Interviews of enablers and barriers to participation, this will be undertaken during:

   - T2 with Group A (8.5 Months post-baseline), and

   - T4 with Group B (post group B undertaking the intervention at 17 months from baseline).

3. Medical record review: Quality and Outcomes of diabetes care: This will be undertaken during:

   - T1 - (6 Months Post Baseline Group A & B).

   - T4 - Measure effect of training program (PSN, Online Modules, Onsite Support) on group B post group B undertaking the intervention at 17 months from baseline (data collected on both group A & B).

   - T6 - Sustainability of effect of training program on group A 29 months post-baseline, and group B 12 months post commencement of intervention.

4. Service systems assessment - Health Service Characteristics (through focus groups): This will be undertaken once during the study between months 3 and 12 from the start of the study.

## Data collection and management

**Data collection and management responsibilities.** There are multiple methods of data collection employed in this trial. The types of data to be collected from the AHW/Ps include their demographics, and outcome questionnaires to assess their knowledge, confidence, attitude, skills, and practice. This data will be collected via online RedCap surveys. On entry into the study each participant will be allocated a unique ID (URID) within RedCap.

Data collection is the responsibility of the research staff of Wardliparingga Aboriginal Health Equity, South Australian Health and Medical Research Institute (SAHMRI), under the supervision of the Principal Investigator. The Study Manager, with guidance and support of the Principal Investigator will manage the data. The Principal Investigator is responsible for ensuring the accuracy, completeness, legibility, and timeliness of the data reported. All data will be entered into a computerised database with regular automated backups and will be protected in accordance with the data management policies of Wardliparingga Aboriginal Health Equity, SAHMRI. Checks for accuracy and completeness will be done shortly after data collection. Computer codes and site randomization schedule will be restricted to the Study Manager and Principal Investigator and maintained in a secure server unavailable to the data analyst.

Electronic data will be stored on servers protected by firewalls, security groups and passwords through S Drive. Accessing the S Drive requires a two-step authentication process that comprises of a SAHMRI log in and then individual access to the S Drive folder arranged by the SAHMRI IT. The SAHMRI IT department has the expertise and resources to support the management of research data to levels required by external data custodians and as outlined in the National Statement. SAHMRI IT requires evidence of ethics approval for those individuals who request access to the study folder located on S Drive. Data are protected in accordance with the data management policies of Wardliparingga Aboriginal Health Equity, SAHMRI.

**Study records retention.** RedCap is an electronic data capture program that will be used to capture most data in this project. RedCap sits on a secure server in Australia that is managed and controlled by SAHMRI. The RedCap unique ID and participant name is stored separately to the participant data which will only record the unique ID. Only the unique ID can be seen and extracted by the research team.

All data will be saved in electronic format on password-protected computers in a locked office of Aboriginal Health Equity, Level 4, North Terrace, SAHMRI. On S Drive one master folder will be created that will hold two folders for: 1) individual level data demographics, primary outcomes, potential confounding and peer support network evaluation variables and enablers and barrier to participation, sustainability and effectiveness, and 2) service level data quality and outcomes of diabetes care and service characteristics. Hardcopy data will be scanned and saved on the S Drive. Audio-transcriptions will be saved on the S Drive.

The Principal Investigator is responsible for retaining data. All data will be stored for 7 years, after which time it will be destroyed in accordance with SAHMRI policy and procedures. Hardcopy data will be destroyed once this is completed at the first opportunity after data collection. Once semi-structured interview data has been transcribed, the audio recording will be destroyed.

## Ethical considerations

**Ethical review.** The study will be conducted in full conformance with principles of the "Declaration of Helsinki" [24], Good Clinical Practice (GCP) [25], the National Statement on Ethical Conduct in Human Research (NHMRC, 2007) [26], Australian Code for the Responsible Conduct of Research (2007) and within the laws and regulations in Australia. As well as, National Health and Medical Research Council, Ethical conduct in research with Aboriginal and Torres Strait Islander Peoples and communities: Guidelines for researchers and stakeholders [27]. The study has ethical approval from the following HRECs:

- SA Department of Health and Wellbeing Human Research Ethics Committee, approval number (2021/HRE00334) in April 2023.

- SA Aboriginal Health Research Ethics Committee (AHREC), approval number (04-21-969) in May 2022.

- In accordance with the *SA Health Research Governance Policy Directive*, Site Specific Assessment (SSA) Approval will be sought from individual public health sites where the project is being conducted.

- A detailed research protocol submitted to ethics is included in the (S7 File. research protocol).

*Inclusivity in global research.* Additional information regarding the ethical, cultural, and scientific considerations specific to inclusivity in global research is included in the (S8 File. Inclusivity in global research questionnaire).

**Confidentiality and De-identification.**

1. *Training program participants.* On entry into the study each participant will be allocated a unique ID (URID) within RedCap. The Study Manager will enter enrolled participant details into RedCap including participant first name, surname, date of birth, email address, enrolment date, consent status and consent form. This will create a participant list and will assign each participant a URID within RedCap, so that their baseline and subsequent evaluations can be linked for the duration of the study. Only the individual who receives the

email link to the RedCap survey can access the survey. Submitted RedCap surveys cannot be accessed by participants at any time through the survey link. RedCap keeps the URID, and participant name separate to the participant data. The participant data will only include the URID. Only de-identified data will be analysed during the study. Individual data will not become part of any report or publication of the project.

2. *Interviews for enablers and barriers.* A subgroup of participants will be interviewed. These interviews will be audio recorded and later transcribed. Interview data will be de-identified by redacting identifiable information in the transcript and will be saved on password protected computers in the office of Aboriginal Health Equity, SAHMRI according to the data retention policy. Data collected in the study will only be accessed by the research team involved in this trial. Data collected by semi-structured interviews with AHW/Ps and multi-disciplinary health care providers will not be linked to the survey data. This data will be aggregated and thematically analysed. Name or date of birth will not be collected from interview participants.

3. *Service systems assessment focus groups.* Each health service will be given a unique ID by the Study Manager. A linking key that holds the service name and unique service ID will be kept in a separate file location. All service level information will include the unique ID only.

4. *Quality of care and patient outcomes (medical record review).* The health service will allocate a unique ID to each patient record, and this will be kept by the service. Only de-identified audit data will be provided to the research team. Should the research team need to clarify anything, they will provide the relevant health service with the unique ID number; the health service will then follow-up the query using the patient record.

Patient outcomes data and training participant data will not be linked. In the health services where the majority of the eligible workforce participated in the training program, it would be likely to expect changes within these services and an improvement in the quality of care provided at the population level. While, sites with a small proportion of their workforce participating, it would be unlikely to expect change at a population level in diabetes management. In both scenarios, the direct impact of the training program on patient outcomes related to diabetes management and the quality of care will not be clearly established. The review of patient outcomes was a request by some services in the co-design of the program.

**Informed consent process.** Participation in the study is voluntary. Several options to ensure potential participants have sufficient information and adequate understanding of the proposed research and the implications for participation will be made available, as follows:

- Participant information sheet,

- Presentations on the study by the research team that may be attended by one or more staff from the same service,

- Individual phone conversation about what it would mean to participate in the study in the way of purpose, method, demands, risks and potential benefits.

With each of these options, potential participants will be invited to have a conversation with a member individually about what participation requires and an opportunity to ask questions. It will be made clear in all options that not participating will not affect their relationship with their employer or the study team or involvement in future research. Participants will not be reimbursed for their time spent undertaking the training program, and at any time participants may withdraw from the study without any penalty. A reason for not participating is not

required. Potential participants themselves will have the ability to provide informed consent, no other person or statutory body is required to decide on their participation in the study.

**Safety considerations.** It is anticipated that any risks to participants will be minimal and limited to discomfort or inconvenience. All risks will be brought to the attention of the Principal Investigator by the research team member/s and/or a participant/s and/or steering committee member. The Principal Investigator will assess if the risk needs to be elevated to the research investigators based on its severity. A risk register will be kept that identifies the risk and severity of the risk, who was affected (not name but if they were a participant, service manager, etc.), who the risk was reported to, the immediate action and how the same risk in the future will be minimized or mitigated. It is the responsibility of the individual who raises the risk with the Principal Investigator to complete the risk register or the Principal Investigator if a participant or someone other than a study team member. The risk register will be included in the papers of each Investigator meeting. Obtaining consent from participants to check on their wellbeing will be discussed in the consent process and is included in consent forms and the distress protocols.

In case of any conflicts between participants during focus groups discussion, which could lead to participants feeling upset or uncomfortable in contributing to discussion. In the event this occurs, the Facilitator will follow a distress protocol which may include pausing or terminating the session. The Facilitator will notify the Principal Investigator of the incident and document the incident. A follow up wellness check will be conducted with the affected participant (s). If conflict arises between the Facilitator and participating staff during the focus group, the Facilitator will terminate the session and contact the Principal Investigator and Service Manager for support. A distress protocol will be followed if distress should arise at any stage during the study.

**Protocol deviations.** The Principal Investigator with all Investigators will monitor protocol deviations and/or adverse events. These will be reported to the SA Department for Health and Wellbeing HREC and site-specific research governance officers within 72 hours of identification of the event. Corrective actions will be implemented promptly.

*Unexpected or serious adverse events*. It is anticipated that the risks associated with participating in this research project will be minimal and limited to discomfort or inconvenience. The Principal Investigator will use continuous vigilance to identify and report adverse events within 72 hours of identification of the event to all approving HRECs and relevant Research Governance Officers.

*Potential confounding variables*. A Directed Acyclic Graphs will be used to determine variables that need to be collected to adjust for potential confounding within the statistical models, this model is used for the ascertainment and controlling of a priori identified confounders of the direct effect of the intervention. Using this conceptual method to inform statistical models, provides additional rigour to control for confounders and to measure the effects of the intervention within the analysis.

Inclusion of participants with a higher starting level of knowledge in the training program could certainly influence the results in various ways; participants with a higher baseline knowledge could affect the overall evaluation of the project, as they might show less improvement than their peers throughout the training program. However, this could have a positive effect for peer support network and onsite practice support, as they will share their experiences and knowledge with their peers. This could lead to skewed perception of the interventions' effectiveness. The main aim is to recruit all interested participants who have completed or currently enrolled in certificate III/IV in Indigenous primary health care, and there will be individual evaluation for all participants to complete at certain timepoints to establish baseline status.

All training program participants are eligible for an interview on the enablers and barriers to participation, effectiveness and sustainability of the training program. These interviews are one component of the training program evaluation. The overall evaluation will seek to understand the effectiveness of the training program on knowledge, confidence, attitude, skills, and practice of Aboriginal Primary Health Care Service providers in providing care to clients with diabetes. However, including only participants who completed the training program for the interview, might introduce bias, therefore, a sensitivity test could be undertaken. As part of this, the research team will invite all participants who have completed or did not complete the training program for an interview.

**Data monitoring.** A project Steering Committee comprising of one representative from each service who is not a study participant will be convened for one hour every two months to oversee the implementation of the study. The study Investigator Group will continue to meet for one hour every 6 weeks to oversee the research conduct and rigor. The Study Manager, with guidance and support of the Principal Investigator will manage the data. All data will be entered into a computerized database with regular automated backups. Checks for accuracy and completeness will be done shortly after data collection. Computer codes and site randomization schedule will be restricted to the Study Manager and Principal Investigator and maintained in a secure server unavailable to the data analyst.

## Statistical analysis

### Quantitative analysis

An epidemiologist, who is a Chief Investigator on the project has led the development of the statistical plan and will undertake the statistical analysis. A descriptive pre-post analysis of survey results due to participating in the PSN alone in the first six months will be calculated using appropriate parametric or non-parametric tests. These analyses will occur after each collection timepoint. In the case of participants who drop-out of the study, any data collected will be analysed by intention to treat. For the PSN, knowledge, attitude and practice of providers will be summarized using descriptive analyses. Differences between pre and post survey results will be calculated using appropriate parametric or non-parametric tests. Provision of evidence-based care and patient biomedical outcomes will be summarized using counts, means and percentages and differences between time collection points reported. Directed Acyclic Graphs (DAGs) are being used to determine variables that need to be collected to adjust for potential confounding within statistical models. DAGs enable the conceptualisation of causal pathways. A DAG has been completed for each primary outcome. In addition, Generalized Linear Mixed-effects Models (GLMMs) are used to deal with outcomes with non-normal distributions, specifically to handle binary outcomes. GLMMs account for correlation structure within health care center (cluster) using a logit or log link for binary outcomes such as prevalence resulting in mixed effect logistic regression, while log link will be used for count outcomes resulting in mixed effects negative binomial regression. Effect measures will also be presented through calculation of number needed to treat and its associated 95% confidence interval. As there is limited evidence on the relationship between participation rate in networks and effect on learning outcomes, we will apply an arbitrary percentage. The minimum participation rate in in PSN sessions in order to be considered as an active participant will be 50% of all the total sessions.

### Qualitative analysis

The qualitative data from the study will be analyzed using a data-driven, inductive, thematic analysis based on the Braun and Clarke's guide for thematic analysis [28]. Inductive analysis is

a process of data coding without fitting it into any pre-existing coding framework. The majority of qualitative data for the study will come from the interviews. The interview transcripts will be audio recorded and transcribed verbatim. The analysis will begin by the analyst repeatedly reading and re-reading the data to become familiar with it. The more formal coding will start once the analyst becomes familiarized with the data. Once the data has been coded, these codes will be sorted into potential themes and subthemes. Once the themes are reviewed, refined, and finalized they will be used to write the final report. We will aim to analyze and report the data separately for AHW and multi-disciplinary team. Combining all disciplines, we will aim to analyze the data by metro and remote.

## Results and discussion

This study presents a unique opportunity to implement a training program for multiple disciplines working in Aboriginal primary health care services and providing diabetes care to Aboriginal clients. With statewide participation of health professionals, the training program has the ability to create baseline knowledge, attitude, practice, skill and confidence with regards to managing type 2 diabetes across primary health care services in SA. As diabetes is so prevalent within the Aboriginal community, having a minimum standard of care provided can benefit those with diabetes and their families. The final reported findings will be published in 2027.

### Dissemination plan

Results will be disseminated via email or via the Teams site in the case of the Peers Support Network evaluation and communicated through the Steering Committee as follow:

- Training program evaluation: Where possible, participating AHW/Ps and Aboriginal primary care health services will receive results as they are produced.

- Peer Support Network: Each month, aggregated results of the survey evaluation will be provided to the Peers Support Network and used to improve the Network.

- Systems assessment: Each service will receive their systems assessment within 3 months of it being completed.

- Medical record review: Each health service will be offered aggregated analyzed service level data on patients of their service only. All services in the study will receive combined aggregated results of the audits as they become available.

Key stakeholders will be provided study results, they include National Association of Aboriginal and Torres Strait Islander Health Workers and Practitioners, Diabetes Australia and their state and territory affiliate members, the National Association of Diabetes Centers, Australian Diabetes Educator Association and the National Aboriginal Community Controlled Health Organization (NACCHO). As host of the E-Learning modules, Diabetes Queensland will be provided aggregated results (only those published) of the evaluations built into the E-Learning modules to inform continual improvements of the E-Learning modules.

### Conclusion

Implementation of a locally co-designed training program will bring together a peer support network of health worker and practitioners working within an Aboriginal primary health setting across south Australia with the intention of evaluating its effectiveness. The results of this study have direct benefit for participants in professional development relating to type 2

diabetes management and in the short-term will contribute to a sparse evidence-base on effectiveness of such training programs.

## Supporting information

**S1 File. Spirit checklist.**
(DOC)

**S2 File. Participants information sheet.**
(DOCX)

**S3 File. Consent form.**
(DOCX)

**S4 File. Peer Support Network guide.**
(DOCX)

**S5 File. Onsite Practice Support guide.**
(DOCX)

**S6 File. Data collection methods.**
(DOCX)

**S7 File. Research protocol.**
(PDF)

**S8 File. Inclusivity in global research questionnaire.**
(DOCX)

**S1 Checklist. Human participants research checklist.**
(DOCX)

## Author Contributions

**Conceptualization:** Odette Pearson, Saravana Kumar, David Jesudason, Paul Zimmet, Gloria C. Mejia, Gary Wittert, Sara Jones, Jane Giles, Natalie Wischer, Sophia Zoungas, Alex Brown, Kim Morey.

**Funding acquisition:** Odette Pearson, Saravana Kumar, David Jesudason, Paul Zimmet, Gloria C. Mejia, Gary Wittert, Sara Jones, Jane Giles, Natalie Wischer, Sophia Zoungas, Alex Brown, Kim Morey.

**Investigation:** Odette Pearson, Saravana Kumar, David Jesudason, Paul Zimmet, Gloria C. Mejia, Gary Wittert, Sara Jones, Jane Giles, Natalie Wischer, Sophia Zoungas, Alex Brown, Kim Morey.

**Methodology:** Odette Pearson, Sana Ishaque, Saravana Kumar, David Jesudason, Paul Zimmet, Gloria C. Mejia, Gary Wittert, Sara Jones, Jane Giles, Natalie Wischer, Sophia Zoungas, Sarah Davey, Tinarra Toohey, Satinder Kaur, Alex Brown, Kim Morey.

**Project administration:** Odette Pearson, Sana Ishaque, Sarah Crossing.

**Supervision:** Odette Pearson.

**Writing – original draft:** Odette Pearson, Sana Ishaque, Sarah Crossing.

**Writing – review & editing:** Odette Pearson, Saravana Kumar, David Jesudason, Paul Zimmet, Gloria C. Mejia, Gary Wittert, Sara Jones, Jane Giles, Natalie Wischer, Sophia Zoungas, Alex Brown, Tina Brodie, Shwikar Othman, Kim Morey.

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
