## [Decision Letter · Decision Letter 0]

16 Jan 2024

PONE-D-23-36718Translation of culturally and contextually informed diabetes training for Aboriginal primary health care providers on Aboriginal client outcomes: Protocol of a cluster randomized trial of effectivenessPLOS ONE

Dear Dr. Pearson,

Thank you for submitting your manuscript to PLOS ONE. After careful consideration, we feel that it has merit but does not fully meet PLOS ONE’s publication criteria as it currently stands. Therefore, we invite you to submit a revised version of the manuscript that addresses the points raised during the review process.

We look forward to receiving your revised manuscript.

Kind regards,

Inge Roggen, M.D., Ph.D.

Academic Editor

PLOS ONE

Journal Requirements:

"Odette Pearson, Alex Brown, David Jesudason, Paul Zimmet, Saravana Kumar, Gloria Mejia, Gary Wittert, Sara Jones, Jane Giles, Natalie Wischer, Sophia Zoungas, and Kim Morey received fund from National Health and Medical Research Council - Medical Research Future Fund (MRFF) Primary Health Care Research Initiative (PHCRI) [APP1200314]. The sponsor does not have any role in the study design, data collection and analysis, decision to publish, or preparation of the manuscript."

Reviewers' comments:

Reviewer's Responses to Questions

**Comments to the Author**

1. Does the manuscript provide a valid rationale for the proposed study, with clearly identified and justified research questions?

Reviewer #1: Yes

Reviewer #2: Yes

Reviewer #3: Yes

Reviewer #4: Yes

2. Is the protocol technically sound and planned in a manner that will lead to a meaningful outcome and allow testing the stated hypotheses?

Reviewer #1: Partly

Reviewer #2: Yes

Reviewer #3: Partly

Reviewer #4: Yes

3. Is the methodology feasible and described in sufficient detail to allow the work to be replicable?

Reviewer #1: Yes

Reviewer #2: Yes

Reviewer #3: Yes

Reviewer #4: Yes

4. Have the authors described where all data underlying the findings will be made available when the study is complete?

Reviewer #1: Yes

Reviewer #2: Yes

Reviewer #3: No

Reviewer #4: Yes

5. Is the manuscript presented in an intelligible fashion and written in standard English?

Reviewer #1: Yes

Reviewer #2: Yes

Reviewer #3: Yes

Reviewer #4: Yes

6. Review Comments to the Author

You may also provide optional suggestions and comments to authors that they might find helpful in planning their study.

Reviewer #1: Major Revision Required

As I understand it, the proposed trial design is summarised below. Right or wrong it would be very helpful in understanding this protocol if a schema of the design along the following lines is included in the Protocol itself.

Period I Period II

Stage 1 X 2 First Evaluation 3 4 Overall Evaluation

Duration 6 months 10 weeks 6 months 10 weeks

A PSN X PSN + E-L Washout PSN

Randomisation

B PSN X PSN Washout PSN+ E-L

The above schema highlights one feature of the design that should be altered as randomisation should occur at the point when the intervention for the two groups diverges (marked at Stage X above).

Also, drawing randomisation from a hat is not regarded as an acceptable form of randomisation. An appropriate computer-based list should be drawn up by the statistical team which (perhaps) could be stratified by the size of the participating sites (the clusters).

Although this comment should be considered by the investigating team, I wonder whether the second period of the design is really worthwhile. Suppose, E-L is not an improvement over PSN alone in Period I, then there seems little point in carrying on into Period II. Obviously, if E-L does improve things then that might be different. So, my question is: Should the trial team consider establishing an independent Data Monitoring Committee to assess whether or not Period II should be implemented? This committee could be presented with a confidential analysis of the data from Period I to enable them to assess the situation and recommend any action (Note, this proposal is different from Data Monitoring as described on Page 21).

I too was concerned that the statistical analysis plan is over complex. Any analysis should be readily understandable to the non-statistical members of the investigating team and the readership of the eventual trial publication. I rather suspect, for example, that on Page 12 “Analysis after the 12-month intervention follow-up is planned, spending alpha equal to 0.005”. will not be readily understood. Is it really necessary?

Summary

This trial seems a very worthwhile project to undertake and the Protocol has been very carefully prepared. However, the design fault with respect to when, and how, randomisation is undertaken itself needs be altered.

Reviewer #2: This study outlines a protocol for a Cluster Randomised Crossover Control Trial to be implemented in Aboriginal primary health care services in South Australia. The trial aims to assess the impact of a culturally and contextually tailored Aboriginal Diabetes Workforce Training Program on the knowledge, attitudes, confidence, skills, and practices of the Aboriginal primary health care workforce concerning diabetes care. The clarity of the study's objectives is commendable. The methodology is robust and appropriate, promising findings with practical implications. Consequently, publication in PLOS ONE is recommended.

Reviewer #3: There are some aspects of the study protocol which require further information in order to answer 'Yes' to the PLoS One reviewer questions. For instance, the protocol does not mention data availability once the study is complete (PLoS reviewer Q4). With regards to Q2 (meaningful outcome and testing of stated hypothesis) I have the following concerns:

1. Sample size for training program. Participation is voluntary and contact with the study team needs to be made by potential participants. The authors state there will be a minimum of 2 participants per site, from 10 potential Aboriginal Community Controlled Health Organization sites. Is there a strategy in case where there are fewer than 2 participants from a site, or several sites with no interested participants? It has been suggested the minimum sample size will be 20, but there is no guarantee that all sites will participate. Can the authors indicate how many of the potential sites are urban, rural and remote as this is proposed comparison in the analysis? It is difficult to assess the applicability of the statistical methods without a sense of the expected sample size.

2. Patient outcomes. It appears that medical records for all diabetes patients at each participating health service will be reviewed, but a yet-to-be determined proportion of practitioners will participate in the training. Will the researchers be able to link patients to specific health practitioners to look for changes in practice? If not, the authors should discuss how likely it is they will see an effect of the intervention if only a small number of practitioners at a site participate.

3. There is no mention of drop-out from the study. How will data for participants who do not complete the online training be treated? Is there a minimum number of PSN sessions a participant needs to attend to be considered as an 'active' participant?

4. The description of the statistical analysis could be more specific about the hypotheses being tested. For instance, the GLMM is proposed for the 12-month post-intervention analysis, but it is not clear what the hypothesis is. Pre-post analysis of PSN survey results is mentioned using parametric or non-parametric tests. Is there any reason why the GLMM could not be used so the clustering effect is accounted for? I would also expect the knowledge, skills and confidence for an individual may fluctuate over time. How is this being addressed, or are the researchers only interested in comparing results at two specific time points in any one analysis?

Other comments:

5. Abstract and discussion says a final report will be produced in 2025. I am not sure how this can occur when the study will be conducted between Feb 2024 and September 2026, with T6 29 months post-baseline.

6. Training program eligibility. It appears that practitioners who have a Graduate Certificate of Diabetes Education and Management are eligible for the intervention. The authors could discuss how inclusion of participants with a higher starting level of knowledge may impact results.

7. Eligibility for interviews about enablers and barriers. Have the authors considered the potential bias introduced if only participants who complete the training are eligible for interview? Participants who do not complete the training may present a unique set of barriers not captured by those who complete.

8. Top of page 14. T2, T4 and T6 are used prior to their explanation.

9. Page 14. “The secondary outcomes for this study relate to the patients care received and patients’ clinical outcomes”. However, the first three dot points that follow focus on the practitioner or health service, not the patient.

10. Page 21. What potential serious adverse events do the authors envisage could develop?

Reviewer #4: It is a well written protocol. Minor language editing is required. A clarification on the working conditions are required. Will the health workers or service providers be able to provide the services after training? It will be good if you give some clarification on why and how trainings will help the staff to address issues of diabetic patients.

Diabetes requires self-care by the patients and family members. Though self-management is mentioned but would be good if you add a module on training staff on how to train patients on tailored self-care.

Best wishes

7. PLOS authors have the option to publish the peer review history of their article (what does this mean?). If published, this will include your full peer review and any attached files.

Reviewer #1: No

Reviewer #2: No

Reviewer #3: No

Reviewer #4: **Yes: **I have done myself. Manmeet Kaur

---

## [Author Response · Author response to Decision Letter 0]

8 Mar 2024

Feedback items Authors response (Amended/added) Page and line number

Journal requirements (Editor feedback)

• Thank you for your feedback; the manuscript has been checked and amended, including file naming and formatting of authors’ affiliations. Pages1-2, 6, 11-12, 16, 20 and 29

"Odette Pearson, Alex Brown, David Jesudason, Paul Zimmet, Saravana Kumar, Gloria Mejia, Gary Wittert, Sara Jones, Jane Giles, Natalie Wischer, Sophia Zoungas, and Kim Morey received fund from National Health and Medical Research Council - Medical Research Future Fund (MRFF) Primary Health Care Research Initiative (PHCRI) [APP1200314]. The sponsor does not have any role in the study design, data collection and analysis, decision to publish, or preparation of the manuscript."

• Thank you, the competing interest section in the manuscript has now been amended to include “This does not alter our adherence to PLOS ONE policies on sharing data and materials.” as an adherence to PLOS ONE polices. Page 27

• The cover letter has also been updated to include the competing interest section. Cover letter document

• This manuscript is a study protocol with no previous pilot data; therefore, a dissemination plan of the results is included on page 25. 

• Data availability statement is provided with in the manuscript “This study applies the principles of Indigenous Data Sovereignty. Ethics approval and consent processes do not include approval for sharing the dataset. The study has ethics approval from SA Department of Health and Wellbeing Human Research Ethics Committee, approval number (2021/HRE00334) in April 2023. In addition, SA Aboriginal Health Research Ethics Committee (AHREC), approval number (04-21-969) in May 2022.” 

The contact details: 

SA department for health and wellbeing research office: Phone: (08) 7117 6635 HealthHumanResearchEthicsCommittee@sa.gov.au

Aboriginal Health Council of South Australia Ltd Tel: (08) 8273 7200 Email: ahcsa@ahcsa.org.au

Page 25, Page 26

Reviewer #1:

As I understand it, the proposed trial design is summarised below. Right or wrong it would be very helpful in understanding this protocol if a schema of the design along the following lines is included in the Protocol itself.

Period I Period II

Stage 1 X 2 First Evaluation 3 4 Overall Evaluation

Duration 6 months 10 weeks 6 months 10 weeks

A PSN X PSN + E-L Washout PSN

Randomisation

B PSN X PSN Washout PSN+ E-L

The above schema highlights one feature of the design that should be altered as randomisation should occur at the point when the intervention for the two groups diverges (marked at Stage X above).

Also, drawing randomisation from a hat is not regarded as an acceptable form of randomisation. An appropriate computer-based list should be drawn up by the statistical team which (perhaps) could be stratified by the size of the participating sites (the clusters).

Although this comment should be considered by the investigating team, I wonder whether the second period of the design is really worthwhile. Suppose, E-L is not an improvement over PSN alone in Period I, then there seems little point in carrying on into Period II. Obviously, if E-L does improve things then that might be different. So, my question is: Should the trial team consider establishing an independent Data Monitoring Committee to assess whether or not Period II should be implemented? This committee could be presented with a confidential analysis of the data from Period I to enable them to assess the situation and recommend any action (Note, this proposal is different from Data Monitoring as described on Page 21).

I too was concerned that the statistical analysis plan is over complex. Any analysis should be readily understandable to the non-statistical members of the investigating team and the readership of the eventual trial publication. 

I rather suspect, for example, that on Page 12 “Analysis after the 12-month intervention follow-up is planned, spending alpha equal to 0.005”. will not be readily understood. Is it really necessary?

Summary

This trial seems a very worthwhile project to undertake and the Protocol has been very carefully prepared. However, the design fault with respect to when, and how, randomisation is undertaken itself needs be altered. 

RESPONSE TO REVIEWER 1

• Thank you for your valuable feedback. The research investigators have agreed that the randomisation can occur at six months. The manuscript text has been amended to include “Randomisation will be undertaken after six months of providing PSN for all enrolled health service sites.” Page 10

• The manuscript text has been amended “All enrolled health service sites will be listed, and a statistician will be responsible for randomizing the health service sites into cluster one (Group A participants) or cluster two (Group B participants) using a block size through a computer program for randomisation.” Page 10

• We would like to thank the reviewer for the comment. The training program was co-designed with seven Aboriginal health primary health services in South Australia, inclusive mainly of Aboriginal health practitioners and workers. It was imperative in the co-design process that all participants had the opportunity to do the training program. In the event that the hypothesis is verified as not effective for group A, a preliminary finding will be reported back to the project Steering Committee Members seeking a collaborative decision on the continuation of the training program for group B. 

• The statistical analysis plan has been reworded to include: A statistician, who is a Chief Investigator on the project has led the development of the statistical plan and will undertake the statistical analysis. A descriptive pre-post analysis of survey results due to participating in the PSN alone in the first six months will be calculated using appropriate parametric or non-parametric tests. These analyses will occur after each collection timepoint. In the case of participants who drop-out of the study, any data collected will be analysed by intention to treat. For the PSN, knowledge, attitude and practice of providers will be summarized using descriptive analyses. Differences between pre and post survey results will be calculated using appropriate parametric or non-parametric tests. Provision of evidence-based care and patient biomedical outcomes will be summarized using counts, means and percentages and differences between time collection points reported. Pages 23-24

• Directed Acyclic Graphs (DAGs) are being used to determine variables that need to be collected to adjust for potential confounding within statistical models. DAGs enable the conceptualisation of causal pathways. A DAG has been completed for each primary outcome. In addition, Generalized Linear Mixed-effects Models (GLMMs) are used to deal with outcomes with non-normal distributions, specifically to handle binary outcomes. GLMMs account for correlation structure within health care center (cluster) using a logit or log link for binary outcomes such as prevalence resulting in mixed effect logistic regression, while log link will be used for count outcomes resulting in mixed effects negative binomial regression. Effect measures will also be presented through calculation of number needed to treat and its associated 95% confidence interval. 

• This sentence has been deleted from the text “Analysis after the 12-month intervention follow-up is planned, spending alpha equal to 0.005”.

Reviewer #2:

This study outlines a protocol for a Cluster Randomised Crossover Control Trial to be implemented in Aboriginal primary health care services in South Australia. The trial aims to assess the impact of a culturally and contextually tailored Aboriginal Diabetes Workforce Training Program on the knowledge, attitudes, confidence, skills, and practices of the Aboriginal primary health care workforce concerning diabetes care. The clarity of the study's objectives is commendable. The methodology is robust and appropriate, promising findings with practical implications. Consequently, publication in PLOS ONE is recommended.

• Thank you for your feedback. 

Reviewer #3 (RESPONSES PROVIDED BY DOT POINT FOR EACH NUMBER):

There are some aspects of the study protocol which require further information in order to answer 'Yes' to the PLoS One reviewer questions. For instance, the protocol does not mention data availability once the study is complete (PLoS reviewer Q4). With regards to Q2 (meaningful outcome and testing of stated hypothesis) I have the following concerns:

1. Sample size for training program. Participation is voluntary and contact with the study team needs to be made by potential participants. The authors state there will be a minimum of 2 participants per site, from 10 potential Aboriginal Community Controlled Health Organization sites. Is there a strategy in case where there are fewer than 2 participants from a site, or several sites with no interested participants? It has been suggested the minimum sample size will be 20, but there is no guarantee that all sites will participate. 

Can the authors indicate how many of the potential sites are urban, rural and remote as this is proposed comparison in the analysis? It is difficult to assess the applicability of the statistical methods without a sense of the expected sample size.

• The manuscript text has been amended to include sample size calculation, drop-out rate and rationales: 

• Sample size calculation and drop-out rate: 

• The study population will be drawn from primary health settings and will include Aboriginal Health Workers and Practitioners, and multidisciplinary health care providers across South Australia. Based on our scoping review “Supporting best practice in the management of chronic diseases in primary health care settings: a scoping review of training programs for Indigenous Health Workers and Practitioners” currently under review, the review reported improvement in participants’ knowledge and confidence of a similar population group between 12.7%-20.97% increase pre-post training, and an increase in confidence in both clinical and non-clinical skills. The sample size for the studies included in this review ranged from 4 to 250 with a total of 1120 participants. The drop-out rate was reported to be between 40 to 70% during the follow up period. Furthermore, based on the publicly available information from the Australian Health Practitioner Regulation Agency (AHPRA), a total of (n=100) Aboriginal health practitioners are currently registered and licenced to work in health settings across South Australia (This does not include Aboriginal Health Workers). There are 11 Aboriginal community-controlled health services and 10 South Australian Local Health Networks, some of which have expressed interest in participating in the study. Overall, we have estimated an uptake of (n=40) and accounted for drop-out rate of 50% of the AHW/P, therefore, (n=20) participants would need to complete the training. This number aligns with previous studies (14-16). Pages (8-9)

• As participation in this project is voluntary, potential participating sites may decline to participate and participants may withdraw from the study. In the event that a participant wishes to withdraw from the study, they will be asked to complete a participant withdrawal form, and this will be reported in the final findings. A minimum of two participants will be recruited from each study site, with no restriction on the maximum number of participants from each service. Recruitment will include participants from different primary health service settings, including urban, rural, and very remote sites. We aim to include a minimum of two sites in each setting. Three months has been allocated to recruit participants. 

• The manuscript text has been amended to include “Recruitment will include participants from primary health service settings, across urban, regional, and remote locations. We aim to include a minimum of two sites in each setting.” Page 9

2. Patient outcomes. It appears that medical records for all diabetes patients at each participating health service will be reviewed, but a yet-to-be determined proportion of practitioners will participate in the training. Will the researchers be able to link patients to specific health practitioners to look for changes in practice? If not, the authors should discuss how likely it is they will see an effect of the intervention if only a small number of practitioners at a site participate.

• In addition to the responses to the previous question related to sample size. 

• Patient outcomes data and training participant data will not be linked. In the health services where the majority of the eligible workforce participated in the training program, it would be likely to expect changes within these services and an improvement in the quality of care provided at the population level. While, sites with a small proportion of their workforce participating, it would be unlikely to expect change at a population level in diabetes management. In both scenarios, the direct impact of the training program on patient outcomes related to diabetes management and the quality of care will not be clearly established. The review of patient outcomes was a request by some services in the co-design of the program. Page 21

3. There is no me

---

## [Decision Letter · Decision Letter 1]

10 Apr 2024

PONE-D-23-36718R1Translation of culturally and contextually informed diabetes training for Aboriginal primary health care providers on Aboriginal client outcomes: Protocol of a cluster randomized trial of effectivenessPLOS ONE

Dear Dr. Pearson,

Thank you for submitting your manuscript to PLOS ONE. After careful consideration, we feel that it has merit but does not fully meet PLOS ONE’s publication criteria as it currently stands. Therefore, we invite you to submit a revised version of the manuscript that addresses the points raised during the review process.

Please correctly address the second reviewers comments from previous review, as they feel their remarks have not been addressed, nor their questions answered.

We look forward to receiving your revised manuscript.

Kind regards,

Inge Roggen, M.D., Ph.D.

Academic Editor

PLOS ONE

Journal Requirements:

Reviewers' comments:

Reviewer's Responses to Questions

**Comments to the Author**

1. Does the manuscript provide a valid rationale for the proposed study, with clearly identified and justified research questions?

Reviewer #1: Yes

Reviewer #4: Yes

2. Is the protocol technically sound and planned in a manner that will lead to a meaningful outcome and allow testing the stated hypotheses?

Reviewer #1: Yes

Reviewer #4: Yes

3. Is the methodology feasible and described in sufficient detail to allow the work to be replicable?

Reviewer #1: Yes

Reviewer #4: Yes

4. Have the authors described where all data underlying the findings will be made available when the study is complete?

Reviewer #1: Yes

Reviewer #4: Yes

5. Is the manuscript presented in an intelligible fashion and written in standard English?

Reviewer #1: Yes

Reviewer #4: No

6. Review Comments to the Author

You may also provide optional suggestions and comments to authors that they might find helpful in planning their study.

Reviewer #1: Accept

The revised paper takes account of my earlier concerns. I wish the investigators good luck with the trial.

Reviewer #4: What ever I mentioned in my previous review none of the comments is addressed. You agree or do not agree to suggestions made needs to be addressed, especially, to the points that you will be able to address and not address and why.

I would strongly suggest language. example for that is in Abstract as well. The last sentence .... health workers outcome? Is it health workers outcome or performance? Outcomes are not the same as performance and misleads the reader. I would suggest the authors to address such issues carefully.

Best wishes

7. PLOS authors have the option to publish the peer review history of their article (what does this mean?). If published, this will include your full peer review and any attached files.

Reviewer #1: No

Reviewer #4: No

---

## [Author Response · Author response to Decision Letter 1]

30 Apr 2024

Dear Reviewers 

Thank you for providing comments on our protocol. We apologise to reviewer 4, we did not purposefully intend to ignore the reviewers comments and hope that we have satisfactorily addressed them in this review to comments. Please find below our comments. 

Journal requirements

Authors response (Amended/added):Thank you for your feedback on the reference lists. 

All references have been cross-checked and amended when required. Pages 27-28

At the introduction section: the first paragraph has been updated with in text citation. Page 4

Reference (2): “Intensive blood-glucose control with sulphonylureas or insulin compared with conventional treatment and risk of complications in patients with type 2 diabetes (UKPDS 33). The Lancet (British edition). 1998;352(9131):837-53.”, was replaced with “Stratton IM, Adler AI, Neil HA, Matthews DR, Manley SE, Cull CA, et al. Association of glycaemia with macrovascular and microvascular complications of type 2 diabetes (UKPDS 35): prospective observational study. Bmj. 2000;321(7258):405-12.” 

Reference (5): Australian Institute of Health and Welfare. The Health and Welfare of Australia’s Aboriginal and Torres Strait Islander Peoples: 2008. Canberra: AIHW; 2008. This reference was replaced with “Australian Institute of Health and Welfare. Aboriginal and Torres Strait Islander Health Performance Framework - Summary report March 2024. AIHW; 2024.” 

Reference (10): “Gibson, O., The impact of primary health care on hospitalisation of Aboriginal and Torres Strait Islander adults with type 2 diabetes in far north Queensland, Australia. 2013, University of South Australia,: Adelaide.” This reference was replaced with “Thomas SL, Zhao Y, Guthridge SL, Wakerman J. The cost-effectiveness of primary care for Indigenous Australians with diabetes living in remote Northern Territory communities. Medical Journal of Australia. 2014;200(11):658-62.” 

Reference (23), was same as reference (15), therefore, it was deleted as a duplicate. “Colleran K, Harding E, Kipp BJ, Zurawski A, MacMillan B, Jelinkova L, et al. Building Capacity to Reduce Disparities in Diabetes: Training Community Health Workers Using an Integrated Distance Learning Model. The Diabetes educator. 2012;38(3):386-96.” 

Reference (25) “Australian Government. Good Clinical Practice (GCP) in Australia. In: The Commonwealth Department of Health and Aged Care, editor. 2020”

This reference was replaced with “Australian Government. Good Clinical Practice (GCP) inspection program Guidance for GCP inspection of clinical trial sites for investigational biologicals and medicinal products. In: The Commonwealth Department of Health and Aged Care, editor. 2022. 

Reviewer #4:

Reviewer feedback: 1- Minor language editing is required. I would strongly suggest language. example for that is in Abstract as well. The last sentence .... health workers outcome? Is it health workers outcome or performance? Outcomes are not the same as performance and misleads the reader. I would suggest the authors to address such issues carefully. 

Authors response (Amended/added)

1-Thank you for your feedback. Health workers outcomes was replaced with “ …health care provider knowledge, attitude, confidence, skill and practice relating to diabetes care”. 

In regards to health care providers outcomes: We are not measuring AHW/Ps performance, we are evaluating the effectiveness of the training program on Aboriginal Health Workers, Practitioners and multidisciplinary team knowledge, confidence, skill, practice and attitude related to diabetes care, and secondary outcomes relating to quality of diabetes care provided and patient outcomes.

Reviewer feedback: 2- A clarification on the working conditions are required. Will the health workers or service providers be able to provide the services after training? It will be good if you give some clarification on why and how trainings will help the staff to address issues of diabetic patients.

Authors response (Amended/added)

2-There will be a follow-up for 12 months following the training program, however, we cannot guarantee that they will continue providing these services after the program finishes. In addition, before the training program finishes, we will follow up with the services and discuss their feedback, and check if they require further workshops/training. 

Why/how the training will help the staff to address issues of diabetes patients? 

Prevalence of self-reported type 2 diabetes among adults Aboriginal and Torres Strait Islander people is estimated of 10.7% in 2018-19 across Australia. It is therefore, expected that about 10.7% of the clients’ medical data at each health service will be included in the medical record review. Therefore, equipping the AHW/Ps with the training program, will help to enhance the quality of care provided to diabetes patients which will leads to improving patients’ health outcome and quality of care at each health service. However, in the health services where the majority of the eligible workforce participated in the training program, it would be likely to expect changes within these services and an improvement in the quality of care provided at the population level. While, sites with a small proportion of their workforce participating, it would be unlikely to expect change at a population level in diabetes management. In both scenarios, the direct impact of the training program on patient outcomes related to diabetes management and the quality of care will not be clearly established. The review of patient outcomes was a request by some services in the co-design of the program. 

The training program has three components (Peer Support Network, E-learning modules and On-Site Support), these components will be accompanied by Systems assessment and Medical record review. 

Upon completion of the training program, an evaluation will be conducted for the participating Aboriginal Health Workers (AHW/Ps). The results of this evaluation will be shared with the Aboriginal primary care health services. 

During the monthly session of the Peer Support Network, the aggregated results of the survey evaluation will be presented to the service. These results will serve as a valuable resource for enhancing the effectiveness of the Network.

For the On-site practice support, participants will be given an opportunity to implement knowledge into practice within their primary health care settings. 

In terms of systems assessment, each health service will receive their individual assessment within three months of its completion. As part of the medical record review each health service will be provided with aggregated and analyzed data specific to their service level. All participating services in the study will receive combined aggregated results of the audits as they become available These aggregated data will help to improve the health service and the participants outcomes/performance on managing diabetes. This comprehensive approach ensures continuous improvement and promotes participants’ attitude toward diabetes management and will provide insight into the effectiveness of the training program on patients’ outcomes. It empowers health services and their health care staff to achieve better health outcomes for their clients. The overall objective of the training program is to evaluate the effectiveness of the training program on Aboriginal Health Workers, Practitioners and multidisciplinary team knowledge, confidence, skill, practice and attitude related to diabetes care, and secondary outcomes relating to quality of diabetes care provided and patient outcomes.

Reviewer feedback: 3- Diabetes requires self-care by the patients and family members. Though self-management is mentioned but would be good if you add a module on training staff on how to train patients on tailored self-care. 

Authors response (Amended/added)

3-Thank you for your valuable feedback. Yes, the e-learning modules are designed to equip the AHW/Ps with the essential knowledge and skills to support the diabetic patients on their self-management skills. The modules will cover a wide range of topics starting from the very basics, moving on to complex aspects of diabetes management. The modules start with a discussion of signs and symptoms, healthy living and ways of diabetes prevention, moving through to learning of Glucose monitoring, how to identify low and high blood glucose levels, and managing medicines and insulins. A module on the diabetes related complications such as Kidney disease, eye disease and foot complications. This module will be followed by support for self-management, where participants learn how to support diabetic patients on managing their diabetes on a daily basis. In addition, participants will acquire more knowledge and understanding on managing diabetes for high risk groups such as pregnancy, elderly and children. Overall, by completing the E-learning Modules participants will be able to support diabetic patients on how to self-manage their diabetes.

---

## [Decision Letter · Decision Letter 2]

31 May 2024

Translation of culturally and contextually informed diabetes training for Aboriginal primary health care providers on Aboriginal client outcomes: Protocol of a cluster randomized trial of effectiveness

PONE-D-23-36718R2

Dear Dr. Pearson,

We’re pleased to inform you that your manuscript has been judged scientifically suitable for publication and will be formally accepted for publication once it meets all outstanding technical requirements.

Kind regards,

Inge Roggen, M.D., Ph.D.

Academic Editor

PLOS ONE

Additional Editor Comments (optional):

Reviewers' comments:

Reviewer's Responses to Questions

**Comments to the Author**

1. Does the manuscript provide a valid rationale for the proposed study, with clearly identified and justified research questions?

Reviewer #4: Yes

2. Is the protocol technically sound and planned in a manner that will lead to a meaningful outcome and allow testing the stated hypotheses?

Reviewer #4: Yes

3. Is the methodology feasible and described in sufficient detail to allow the work to be replicable?

Reviewer #4: Yes

4. Have the authors described where all data underlying the findings will be made available when the study is complete?

Reviewer #4: Yes

5. Is the manuscript presented in an intelligible fashion and written in standard English?

Reviewer #4: Yes

6. Review Comments to the Author

You may also provide optional suggestions and comments to authors that they might find helpful in planning their study.

Reviewer #4: All the points raised in the review have been appropriately addressed. There are no more issues that needs to addressed.

7. PLOS authors have the option to publish the peer review history of their article (what does this mean?). If published, this will include your full peer review and any attached files.

Reviewer #4: **Yes: **Prof. Manmeet kaur

---

## [Editor Report · Acceptance letter]

18 Jun 2024

PONE-D-23-36718R2 

PLOS ONE

Dear Dr. Pearson, 

I'm pleased to inform you that your manuscript has been deemed suitable for publication in PLOS ONE. Congratulations! Your manuscript is now being handed over to our production team.

Kind regards, 

on behalf of

Prof. Inge Roggen 

Academic Editor

PLOS ONE